# Characterization of the Complete Mitochondrial Genome of a Flea Beetle *Luperomorpha xanthodera* (Coleoptera: Chrysomelidae: Galerucinae) and Phylogenetic Analysis

**DOI:** 10.3390/genes14020414

**Published:** 2023-02-04

**Authors:** Jingjing Li, Bin Yan, Hongli He, Xiaoli Xu, Yongying Ruan, Maofa Yang

**Affiliations:** 1Institute of Entomology, Guizhou University, Guiyang 550025, China; 2Guizhou Provincial Key Laboratory for Agricultural Pest Management of the Mountainous Region, Guiyang 550025, China; 3Plant Protection Research Center, Shenzhen Polytechnic, Shenzhen 518055, China

**Keywords:** leaf beetles, mitochondrion, monophyly, next-generation sequencing, phylogeny

## Abstract

In this study, the mitochondrial genome of *Luperomorpha xanthodera* was assembled and annotated, which is a circular DNA molecule including 13 protein-coding genes (PCGs), 22 transfer RNA (tRNA) genes, 2 ribosomal RNA genes (12S rRNA and 16S rRNA), and 1388 bp non-coding regions (A + T rich region), measuring 16,021 bp in length. The nucleotide composition of the mitochondrial genome is 41.3% adenine (A), 38.7% thymine (T), 8.4% guanine (G), and 11.6% cytosine (C). Most of the protein-coding genes presented a typical ATN start codon (ATA, ATT, ATC, ATG), except for ND1, which showed the start codon TTG. Three-quarters of the protein-coding genes showed the complete stop codon TAR (TAA, TAG), except the genes COI, COII, ND4, and ND5, which showed incomplete stop codons (T- or TA-). All the tRNA genes have the typical clover-leaf structure, except tRNA^Ser1^ (AGN), which has a missing dihydrouridine arm (DHU). The phylogenetic results determined by both maximum likelihood and Bayesian inference methods consistently supported the monophyly of the subfamily Galerucinae and revealed that the subtribe Luperina and genus *Monolepta* are polyphyletic groups. Meanwhile, the classification status of the genus *Luperomorpha* is controversial.

## 1. Introduction

The insect mitochondrial DNA (mtDNA) is a double-stranded circular DNA molecule, with the maternal inheritance features of a relatively small molecular mass, a relatively conservative gene arrangement, and a rapid rate of gene evolution, etc. [1]. The mitochondrial genes have been widely used in species identification, the estimation of evolutionary relationships, and the recognition of population structure and phylogeography [2]. The mitogenome contains 13 protein-coding genes (PCGs), 2 ribosomal RNA genes (rRNAs), 22 transfer RNA genes (tRNAs), and 1 A + T-rich region [3]. Nowadays, mitochondrial DNAs (mtDNAs) have been commonly used as molecular markers for reconstructing phylogenetic relationships, revealing population genetic structures, estimating divergence times, identifying relatedness between recently diverged species, etc. [4,5].

The invasive flea beetle, *Luperomorpha xanthodera* (Coleoptera: Chrysomelidae: Galerucinae), was originally described by Fairmaire [6] in Jiangxi Province of China [7]. It has previously been recorded in 16 provinces in China, distributing throughout Jilin, Gansu, Shanxi, Shandong, Jiangsu, Zhejiang, Hubei, Jiangxi, Hunan, Fujian, Taiwan, Guangdong, Guangxi, Sichuan, Guizhou, and Yunnan. It has also been found in many other countries, such as Korea, Japan, Italy, France, the Netherlands, and Poland [8,9,10,11,12]. This flea beetle is a crucial pest of economic crop, such as roses (Rosa) and *Zanthoxylum* spp. Their larvae are commonly found concealed in the roots of plants, while the adults feed on grasses, shrubs, and trees flowers. *Luperomorpha xanthodera* has been reported to feed on 21 different plants, including white mustard (*Sinapis alba* L., Brassicaceae), hyssop (*Hyssopus officinalis* L.), and *Monarda* spp. (Lamiaceae), etc. [8]. To our knowledge, in the natural population of *Luperomorpha xanthodera*, the females constituted the vast majority [13]. This means that this species is highly susceptible to outbreaks.

In the past, the species of the genus *Luperomorpha* Weise were placed among the Alticinae (currently known as Alticini) due to the presence of the metafemoral spring. However, some scholars [14,15] studied the morphological characters of Galerucinae and Alticinae (currently known as Galerucini and Alticini), and they discovered that the metafemoral spring could not be the sole trait that divides the two subfamilies. In a later paper, it was noted that although *Luperomorpha* has a metafemoral spring, it only has one basal patch at the base of the ventral side of the sheath wing and should be transferred to Galerucinae. However, the classification and taxonomic position of the genus *Luperomorpha* has continued to prove problematic. Nowadays, only one species of *Luperomorpha* has been reported for a complete mitogenome.

For the present study, we sequenced and annotated *Luperomorpha xanthodera*’s complete mitogenome and studied its characteristics. This study aimed to acquire the data on mitochondrial genome, additional analyses of the composition of the *Luperomorpha xanthodera* mitochondrial genome, the values of the relative synonymous codon usage (RSCU), the evolutionary rate of *Luperomorpha* and the phylogenetic relationship, etc. We believe such results will be helpful in studies of the population genetics and phylogenetic relationships of this species, as well as studies on other flea beetles in the future. We adopted Nie’s classification system [16] to conduct the subsequent analyses.

## 2. Materials and Methods

### 2.1. Sample Collection and DNA Extraction

The invasive flea beetle, *Luperomorpha xanthodera*, was collected on the *Zanthoxylum planispinum* from Zhenfeng county, Guizhou, China (25°24′ N, 105°35′ E), on 23 June 2021. The total DNA was extracted from the entire body without the abdomen using the DNeasy Blood & Tissue Kit (Cat. No. 69504) (Qiagen, Hilden, Germany), as per the manufacturer’s protocol. The voucher specimen’s genome DNA and male genitalia were deposited at the Institute of Entomology, Guizhou University, Guiyang, China (GUGC). The identification of adult specimens was based on morphological characteristics [8,10].

### 2.2. Genome Sequencing, Assembly and Annotation

Illumina TruSeq libraries with an average insert size of 300 bp were prepared and sequenced on the Illumina NovaSeq 6000 platform (Beijing Berry Bioinformatics Technology Co., Ltd., Beijing, China), obtaining 150 bp paired-end reads. The mitogenome was preliminarily annotated by Mitoz 2.4-α [17], with the invertebrate mitochondrial genetic codes. The MITOS web server (http://mitos.bioinf.uni-leipzig.de/index.py, accessed on 14 December 2021) [18] and the tRNAscan-SE search server (http://lowelab.ucsc.edu/tRNAscan-SE/, accessed on 14 December 2021) [19,20] were used to reconfirm and predict the locations and secondary structures of the tRNA genes, and then the tRNA secondary structures were visualized using Adobe Illustrator. The circular mitogenome maps were visualized using Geneious Prime [21].

### 2.3. Mitogenome Sequence Analyses

MEGA 7.0 [22] was used to obtain the nucleotide composition statistics, relative synonymous codon usage (RSCU) of all the protein-coding genes (PCGs), and the base composition. The following formula was used to calculate the strand asymmetry: AT skew = [A − T]/[A + T] and GC skew = [G − C]/[G + C] [23]. In the A + T rich region, the tandem repeat elements were verified using the Tandem Repeats Finder program (http://tandem.bu.edu/trf/trf.html, accessed on 23 December 2021) [24]. The non-synonymous substitutions (Ka) and synonymous substitutions (Ks) of 13 PCGs of 3 species (*A. zanthoxylumi*, *A. planipennis*, and *A. ornatus*) were calculated using DnaSP v 5 [25].

### 2.4. Phylogenetic Analysis

For the phylogenetic analysis, 24 additional mitogenome sequences were downloaded from GenBank, and we selected *Chrysomela vigintipunctata* and *Chrysolina aeruginosa* as the outgroups (Table 1). The L-INS-I strategy in MAFFT software was used for the sequences of amino acids of 13 PCGs, trimmed using trimAl v1.4.1 [26] with the heuristic method ‘automated1’ to eliminate the gap-only and ambiguous-only locations, and was concatenated using FASconCAT g v1.04 [27].

Using the amino acid sequence with 13 PCGs in the phylogenetic analysis, the maximum likelihood (ML) and Bayesian inference (BI) were used to create phylogenetic trees. The ML analyses were carried out using IQ-TREE V1.6.3 [28] and 1000 ultrafast bootstrap [29] and 1000 SHaLRT replicates [30]. The best model was determined through the AIC and BIC scores. The PMSF model [31] was used for the amino acid sequence matrix with 13 PCGs by specifying the profile mixture model with the option “-mtInv + C60 + FO + R”. Under the CAT + GTR model [32], the amino acid sequence matrix with 13 PCGs was further analyzed with Bayesian inference using PhyloBayes MPI v1.8b [33] and independently run for 10,000 generations for two separate chains. We calculated the largest discrepancy (maxdiff) across all the bipartitions using the bpcomp program. When the maxdiff was <0.3, runs were considered to achieve a good convergence. The tracecomp program was also used to evaluate the discrepancy between two independent runs. The minimum effective size >50 was recognized to be a good run. In this study, values greater than 98 percent (SH-aLRT, UFBoot2) and 0.99 (posterior probability) are considered to be of a “high” support; values of 80 percent to 98 percent for SH-aLRT, 95 percent to 98 percent for UFBoot2, and 0.95 percent to 0.99 percent for the posterior probability are considered to be of a “moderate” support; and values of 95 percent for UFBoot2 and 0.95 percent for the posterior probability are considered to be of a “low” support. The resulting trees were visualized and edited using FigTree v.1.4.2.

## 3. Results and Discussion

### 3.1. Mitogenome Organization and Nucleotide Composition

In this study, the complete mitogenome of *Luperomorpha xanthodera* was successfully obtained and uploaded to GenBank under the accession number ON631248 (Table 1; Figure 1). It has a total length of 16,021 bp and includes 37 genes, including 13 protein-coding genes (PCGs), 2 ribosomal RNA (rRNA) genes, and 22 transfer RNA (tRNA) genes, as well as a noncoding control region (A + T-rich region). The gene order of the *Luperomorpha xanthodera* mitogenome was consistent with the ancestral gene order of *Drosophila yakuba*, this is thought to be the ground pattern for insect mitogenomes. As with most galerucine species, 23 genes were on the majority strand (F-strand), and the other 14 genes (tRNA^Gln^, tRNA^Cys^, tRNA^Tyr^, tRNA^Phe^, ND5, tRNA^His^, ND4, ND4L, tRNA^Pro^, ND1, tRNA^Leu^, 16S rRNA, tRNA^Val^, 12S rRNA) were on the minority strand (R-strand) (Table 2).

The *Luperomorpha xanthodera* mitogenome contained 41 bp overlapping regions in 11 pairs of neighboring genes ranging in length from 1 to 8 bp. The longest overlapping region of 8 bp was located between tRNA^Trp^ and tRNA^Cys^ and tRNA^Tyr^ and COⅠ, respectively. A total of 34 bp intergenic nucleotides (IGN) ranging in size from 1 to 17 bp were distributed across seven locations. The longest intergenic spacer was found between the tRNA^Ser^ and ND1 (Table 2). These overlapping and intergenic regions are very frequent in flea beetle mitochondrial genomes [34,35,36]. The start codons and termination codons of all the PCGs genes coincided with the typical coleopteran [37] (truncated termination codons).

The overall nucleotide composition of *Luperomorpha xanthodera* is 41.3% A, 38.7% T, 11.6% C, and 8.4% G and includes a high A + T content (80%) and positive AT skew (0.032) and negative GC skew values (−0.160), this is comparable to other flea beetles. For instance, all the 27 flea beetles we analyzed had positive AT skews and negative GC skews, indicating that the base compositions of the flea beetle mitogenomes were, overall, biased towards A and C (Table 1). Furthermore, the A + T content of the third codon position was the highest (92.2%), while the A + T content of the second codon position was the lowest (69.0%). The AT skews of all the codon positions are positive, indicating a slightly higher occurrence of a likeness to T nucleotides. Meanwhile, the GC skews of the first codon position are positive, but the other two codon positions are negative.

### 3.2. Protein-Coding Genes and Codon Usage

The PCGs include seven NADH dehydrogenases (ND1-ND6 and ND4L), three cytochrome c oxidases (COⅠ-COⅢ), two ATPases (ATP6 and ATP8), and one cytochrome b (CYTB). The total length of the 13 PCGs of *Luperomorpha xanthodera* was 11,116 bp, with an A + T content of 78.2%. Additionally, nine of its genes (ND2, COⅠ, COⅡ, ATP8, ATP6, COⅢ, ND3, ND6, CYTB) are located on the major strand (F-strand), and the others (ND5, ND4, ND4L, ND1) are located on the minor strand (R-strand). The ND5 gene was the longest at 1705 bp, and the smallest was the ATP8 gene at 156 bp. The base compositions of the 13 PCG genes were as follows: 40.2% A, 38.0% T, 9.4% G, and 12.5% C (Table 3). Most PCGs of *Luperomorpha xanthodera* were initiated with the typical ATN codon (except ND1 with TTG). Most PCGs (ND2, ATP8, ATP6, COⅢ, ND3, ND4L, ND6, CYTB, and ND1) had the complete stop codon of TAA or TAG, whereas four PCGs (COⅠ, COⅡ, ND4, and ND5) ended with the incomplete stop codon (T-). Some scholars believe that incomplete termination codons will be repaired by polyadenylation after transcription [38,39]. Based on 3698 codons, the 13 PCGs’ relative synonymous codon usage (RSCU) values were determined. UUU (Phe), UUA (Leu2), AUU (Ile), and AUA (Met) were the four most common codons. The preferred codons all ended in A or U, contributing to the overall A + T bias of the mitogenomes. (Table 4; Figure 2).

We calculated the nonsynonymous substitution rate (Ka), synonymous substitution rate (Ks), and the ratio of Ka/Ks for each PCG in three species (Figure 3). In all PCGs, the highest Ks value was exhibited by ATP6, whereas the Ka value of ATP8 was distinctly higher than others (Figure 3). The average Ka/Ks ranged from 0.181 (ATP6) to 1.037 (ND4). It is noteworthy that with the exception of ND4, all other PCGs of Ka/Ks values are lower than 1, demonstrating that the mutations were exchanged out by synonymous substitutions (Figure 3). The ND4 gene reached 1.037; this result provides an indication that positive selection had a dominating impact on the evolution of ND4 gene. The smallest Ka/Ks -values were 0.181 for COX1 and ATP6, which was considered to be a strong purifying selection.

### 3.3. Transfer and Ribosomal RNA Genes

The length of the 22 tRNA genes of *Luperomorpha xanthodera* range from 62 bp (tRNA^Ile^) to 70 bp (tRNA^Lys^), comprising 8.94% of the complete mitogenome (Table 2), of which all the tRNA genes can be folded into the typical cloverleaf structure, except for trnS1(AGN), whose dihydrouridine (DHU) arm is converted to a simple loop (Figure 4)—this characteristic is frequent in flea beetle mitogenomes [40]. The anticodon of tRNA^Lys^ mutations is TTT, and when we identified the insects of Chrysomelidae, using this would be one of the quickest methods [16].

The *Luperomorpha xanthodera* of ribosomal RNA (rRNA) is composed of 16S rRNA (situated between tRNA^Leu^ and tRNA^Val^) and 12S rRNA (situated between tRNA^Val^ and the A + T rich region). The size of the two genes was 1280 bp (16S rRNA) and 812 bp (12S rRNA), respectively. The A + T content reached 84.2% in *Luperomorpha xanthodera*. Furthermore, the two rRNAs contained a positive AT skew and negative GC skew (Table 3).

### 3.4. A + T Rich Region

The A + T-rich region is an important noncoding region in insect mitogenome, which regulates the mitochondrial DNA (mtDNA) transcription and replication [41,42,43]. The A + T rich region of *Luperomorpha xanthodera* is located between 12S rRNA and tRNA^Ile^ and is 1388 bp in length (Table 2). Meanwhile, the A + T content was 87.3%, with positive AT skews (0.010) and negative GC skews (−0.213) (Table 3). Of note, we found four tandem repeat units in the control region, ranging from 13 to 35 bp. Furthermore, a poly-A region and three poly-T regions were found in the control region (Figure 5).

### 3.5. Phylogenetic Relationships

In the past two decades, molecular systematic approaches are widely used for disclosing unsettled classification and phylogenetic relationships in Insecta [44,45]. The phylogenetic trees of 37 mitochondrial genes (including one newly generated sequence, 24 other Chrysomelidae sequences from GenBank and two outgroup sequences) were generated from a single dataset (amino acids sequence with the 13 PCGs) using maximum likelihood (ML) and Bayesian inference; the BI tree and ML tree are shown in this paper (Figure 6 and Figure 7). The two phylogenetic trees produced consistent results: (1) the Galerucinae is represented as a monophyletic group; (2) the Alticini were clustered into a monophyletic clade and formed sister groups with other subtribes of the Galerucini; (3) the Luperina was the polyphyletic group, which was consistent with the results of Gillespie et al. [46]; and (4) *Monolepta occifluvis* does not cluster with the other two species of the genus *Monolepta* but is instead sister to *Paleosepharia posticata* with high support values (0.99/98.7/100). These findings are consistent with earlier mitogenome-based research [47], which indicates that *Monolepta* is a polyphyletic genus. There are distinctions between the two trees as well. In the BI tree, the *Hoplasoma unicolor* is recovered as the sister group of the *Oises livida*, and their combined clade is grouped with *Luperomorpha* Weise. In the ML tree, *Luperomorpha* and Luperina showed a closer relationship.

According to our phylogenetic tree, the two *Luperomorpha* species are nested in Galerucini. The cladistic taxonomic position of several genera (including *Luperomorpha* Weise) has been rotating between Galerucini and Alticini in evolutionary studies of Chrysomelidae [48,49,50], and these genera are known as the ‘problematic’ genera [15]. As research on these two tribes has progressed, their taxonomic basis has shifted from a single character analysis (metafemoral spring) to a combination of characters (metafemoral spring, hind wing venation, female spermatheca, male aedeagus, etc.), resulting in *Luperomorpha* Weise being removed from Alticinae (currently Alticini) and added to Galerucinae (currently Galerucini) [51]. Some phylogenetic studies also support the *Luperomorpha* Weise being attributed to Galerucini [16,52].

## 4. Conclusions

At present, we sequenced and analyzed *Luperomopha xanthodera*’s complete mitogenome. The complete mitochondrial genome of *Luperomopha xanthodera* has a final size of 16,021 bp, with 80% AT content. The circular mitogenome encoded a typical set of 37 genes (13 PCGs, two rRNAs, 22 tRNAs, and one control region). The majority of PCGs use ATN (except ND1 with TTG) as their start codon and TAA/TAG/T- as the stop codons. For the analysis of the selective pressure, we discovered that most of the 13 PCGs of *Luperomorpha* were less than one, particularly COX1 and ATP6 genes, which had the lowest value, indicating a high conservation. This revealed that PCGs were subjected to purifying selection in the genus, while the ND4 gene has a high value, demonstrating that it may have been mutated during the evolution process. Phylogenetic trees based on the mitogenomes of 27 species contributed to the scientific classification of *Luperomopha xanthodera*. Our study identified a close relationship between *Luperomopha* Weise and Luperini (currently Luperina). Overall, our results provide a hint for the phylogenetic position of *Luperomorpha*, and they have provided basic genetic information for understanding the phylogeny and evolution of leaf beetles.

## Figures and Tables

**Figure 1 genes-14-00414-f001:**
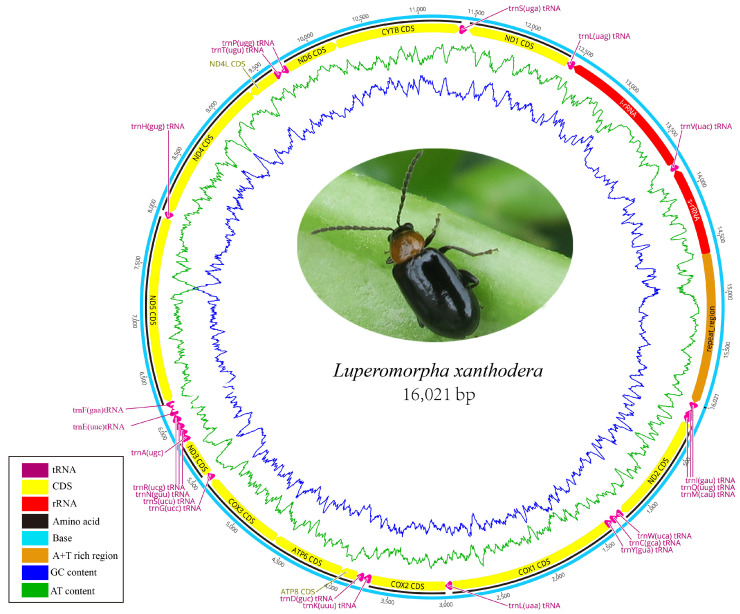
Complete mitogenome map of *Luperomorpha xanthodera*. The majority strand (F-strand) encodes 14 genes, while the minority strand (R-strand) encodes 23 genes. Protein and rRNA genes are denoted using standard abbreviations. tRNA genes are represented by a single letter for each amino acid, with two leucine tRNAs and two serine tRNAs separated by numbers.

**Figure 2 genes-14-00414-f002:**
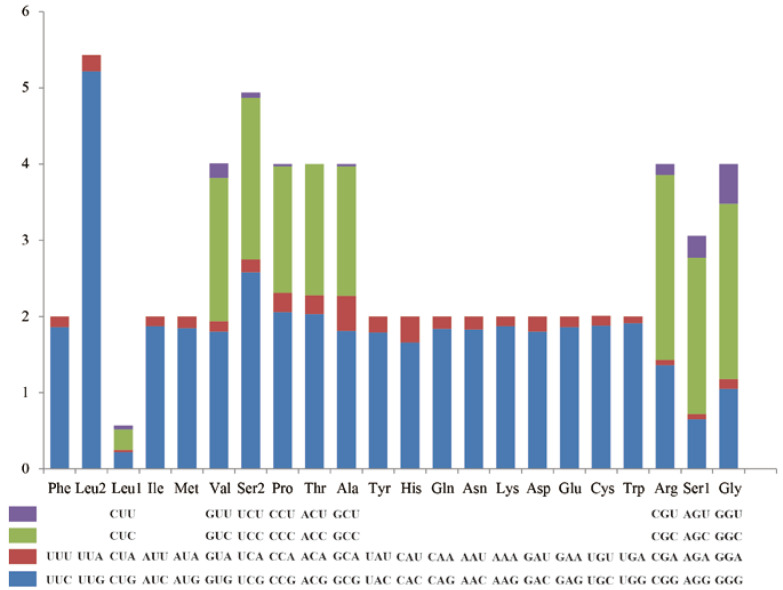
Relative synonymous codon usage (RSCU) of mitogenome of *Luperomorpha xanthodera*. Codon families are depicted in boxes below the *x*-axis; the box colors represent the stacked columns; and the values of RSCU are shown on the *y*-axis.

**Figure 3 genes-14-00414-f003:**
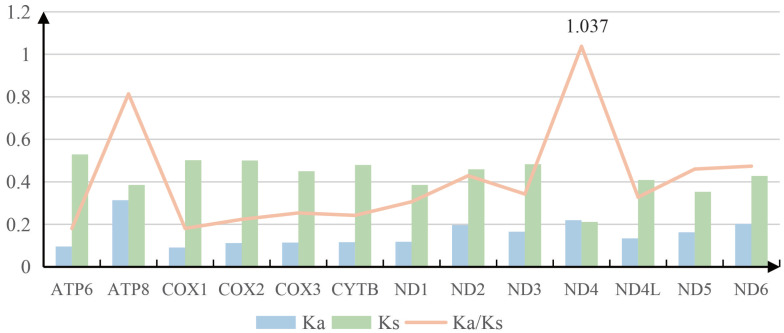
Evolutionary rates of 13 PCGs in between *Luperomorpha xanthodera* and *Luperomorpha hainana*. A blue boxes, green boxes, and orange line indicate the non-synonymous substitutions rate (Ka), the synonymous substitutions rate (Ks), and the ratio of the rate of non-synonymous substitutions to the rate of synonymous substitutions (Ka/Ks), respectively.

**Figure 4 genes-14-00414-f004:**
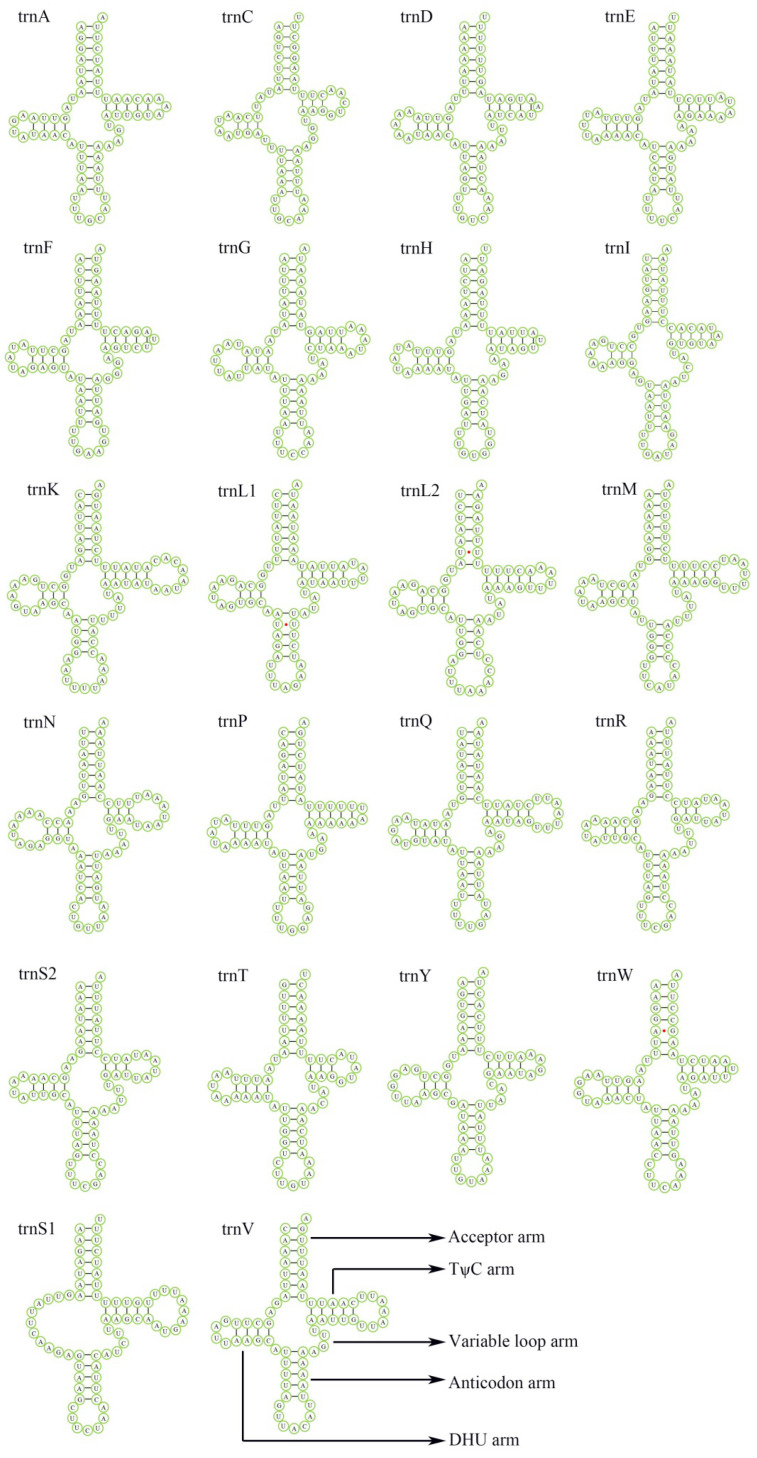
Predicted secondary structures of 22 tRNAs in *Luperomorpha xanthodera*. Lines (-) indicate Watson–Crick base pairings, whereas dots (•) indicate unmatched base pairings. The uppercase letter indicate the identity of each tRNA genes. Structural elements in tRNA arms and loops are illustrated as for trnV.

**Figure 5 genes-14-00414-f005:**
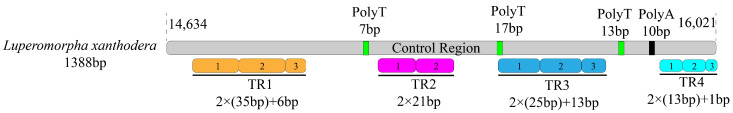
A + T-rich region of *Luperomorpha xanthodera* mitogenome. Gray blocks represent the A + T rich region. The numbers at the beginning and end of the gray blocks show the position of the control region in the *Luperomorpha xanthodera* mitogenome. R stands for repeat unit, and the number denotes the number of repeats. The black and green blocks represent the structures of poly-A and poly-T, respectively.

**Figure 6 genes-14-00414-f006:**
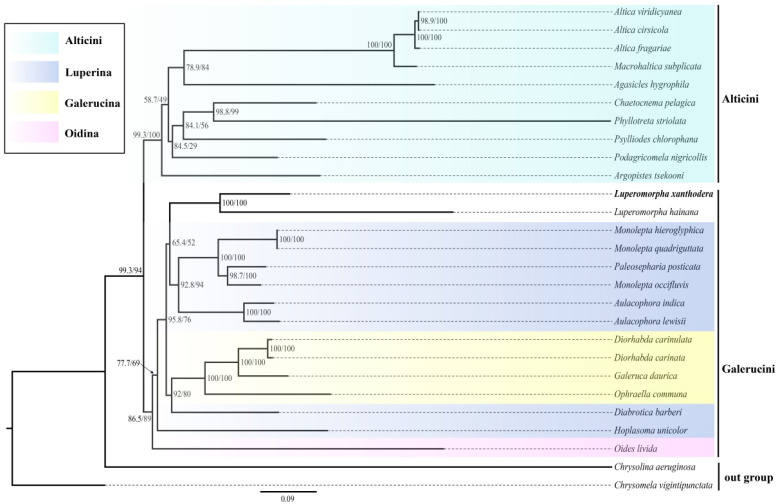
ML analysis was conducted using IQ-TREE with the dataset of amino acids of 13 PCGs. Note: Name in black bold show the phylogenetic position of *Luperomorpha xanthodera* that we sequenced in this research. The numbers on the nodes are the SH-aLRT support (%) and ultrafast bootstrap support (%).

**Figure 7 genes-14-00414-f007:**
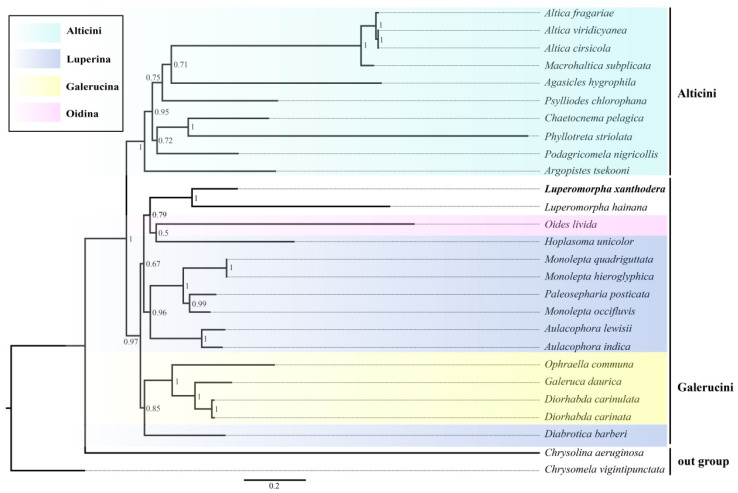
BI analysis was conducted using PhyloBayes with the dataset of amino acids of 13 PCGs. Note: Name in black bold show the phylogenetic position of *Luperomorpha xanthodera* that we sequenced in this research. The numbers on the nodes are the posterior probability.

**Table 1 genes-14-00414-t001:** Mitochondrial genomes were used for the phylogenetic analyses in this research.

Subfamily	Tribe	Species	Length (bp)	G + C%	A + T%	AT-Skew	GC-Skew	GenBank No.
Galerucinae	Alticini	*Agasicles hygrophila*	15,917	24.9	75.2	0.05	−0.19	NC_028332
		*Altica cirsicola*	15,864	22.2	77.8	0.03	−0.21	NC_042876
		*Altica fragariae*	16,220	22.0	78	0.03	−0.22	NC_042875
		*Altica viridicyanea*	16,706	22.3	77.8	0.04	−0.23	NC_048472
		*Argopistes tsekooni*	16,552	20.5	79.5	0.04	−0.19	NC_045929
		*Chaetocnema pelagica*	16,331	23.1	76.9	0.06	−0.20	NC_041170
		*Macrohaltica subplicata*	15,840	22.0	78	0.02	−0.22	NC_041169
		*Phyllotreta striolata*	15,689	24.2	75.8	0.05	−0.26	NC_045901
		*Psylliodes chlorophana*	14,561	23.7	76.3	0.05	−0.22	NC_053362
		*Luperomorpha hainana*	16,081	22.2	77.8	0.05	−0.25	MF960124
		*Luperomorpha xanthodera*	16,021	20	80	0.03	−0.16	ON631248
	Incertae sedis	*Paleosepharia posticata*	15,729	20.2	79.8	0.02	−0.17	NC_033532
		*Podagricomela nigricollis*	16,756	21.3	78.8	0.05	−0.18	NC_041423
	Galerucini	*Diorhabda carinata*	16,232	22.1	77.9	0.03	−0.19	NC_042945
		*Diorhabda carinulata*	16,298	21	78.9	0.03	−0.18	NC_042946
		*Galeruca daurica*	16,615	21.8	78.1	0.04	−0.20	NC_027114
		*Ophraella communa*	16,553	22.6	77.4	0.06	−0.18	NC_039710
	Luperini	*Aulacophora indica*	16,246	21.4	78.6	0.03	−0.20	NC_047467
		*Aulacophora lewisii*	15,691	21.1	78.9	0.03	−0.18	NC_039712
		*Diabrotica barberi*	16,366	20.8	79.2	0.04	−0.18	NC_022935
		*Hoplasoma unicolor*	14,568	22.2	77.8	0.05	−0.23	NC_041168
		*Monolepta hieroglyphica*	15,963	19.7	80.3	0.02	−0.18	NC_057489
		*Monolepta occifluvis*	15,998	20.7	79.2	0.03	−0.20	NC_045838
		*Monolepta quadriguttata*	16,130	19.7	80.3	0.02	−0.19	NC_039711
	Oidini	*Oides livida*	16,127	20.0	79.9	0.05	−0.14	MF960098
Chrysomelinae	Chrysomelini	*Chrysomela vigintipunctata*	17,474	21.7	78.2	0.04	−0.21	NC_050933
		*Chrysolina aeruginosa*	16,335	24.5	75.5	0.07	−0.25	NC_052915

**Table 2 genes-14-00414-t002:** Organization of the *Luperomorpha xanthodera* mitochondrial genome. IGN: intergenic nucleotides. Negative values refer to overlapping nucleotides. dots (•) indicate no value, hyphens (-) represent the size of overlapping regions.

Gene	Direction	Location (bp)		Size (bp)	Codon		Anti-Codon	IGN
		from	to		Start	Stop		
tRNA^Ile^	F	1	62	62	•	•	GAT	
tRNA^Gln^	R	63	131	69	•	•	TTG	0
tRNA^Met^	F	131	199	69	•	•	CAT	-1
ND2	F	200	1210	1011	ATT	TAA	•	0
tRNA^Trp^	F	1211	1274	64	•	•	TCA	0
tRNA^Cys^	R	1267	1328	62	•	•	GCA	-8
tRNA^Tyr^	R	1329	1391	63	•	•	GTA	0
COⅠ	F	1384	2926	1543	ATT	T	•	-8
tRNA^Leu^	F	2927	2991	65	•	•	TAA	0
COⅡ	F	2992	3676	685	ATG	T	•	0
tRNA^Lys^	F	3677	3746	70	•	•	TTT	0
tRNA^Asp^	F	3747	3809	63	•	•	GTC	0
ATP8	F	3810	3965	156	ATC	TAA	•	0
ATP6	F	3959	4633	675	ATG	TAA	•	-7
COⅢ	F	4633	5421	789	ATG	TAA	•	-1
tRNA^Gly^	F	5424	5488	65	•	•	TCC	2
ND3	F	5489	5842	354	ATT	TAG	•	0
tRNA^Ala^	F	5841	5904	64	•	•	TGC	-2
tRNA^Arg^	F	5905	5968	64	•	•	TCG	0
tRNA^Asn^	F	5966	6033	68	•	•	GTT	-3
tRNA^Ser^	F	6034	6100	67	•	•	TCT	0
tRNA^Glu^	F	6102	6165	64	•	•	TTC	1
tRNA^Phe^	R	6165	6227	63	•	•	GAA	-1
ND5	R	6228	7932	1705	ATT	T	•	0
tRNA^His^	R	7933	7995	63	•	•	GTG	0
ND4	R	8005	9325	1321	ATG	T	•	9
ND4L	R	9319	9600	282	ATG	TAA	•	-7
tRNA^Thr^	F	9603	9665	63	•	•	TGT	2
tRNA^Pro^	R	9666	9729	64	•	•	TGG	0
ND6	F	9732	10,235	504	ATT	TAA	•	2
CYTB	F	10,235	11,374	1140	ATG	TAG	•	-1
tRNA^Ser^	F	11,373	11,439	67	•	•	TGA	-2
ND1	R	11,457	12,407	951	TTG	TAG	•	17
tRNA^Leu^	R	12,409	12,473	65	•	•	TAG	1
16S rRNA	R	12,474	13,753	1280	•	•	•	0
tRNA^Val^	R	13,754	13,821	68	•	•	TAC	0
12S rRNA	R	13,822	14,633	812	•	•	•	0
A + T rich region	F	14,634	16,021	1388	•	•	•	0

**Table 3 genes-14-00414-t003:** Nucleotide composition of the *Luperomorpha xanthodera* mitogenomes.

Genes	Size (bp)	Nucleotides Composition						ATskew	GCskew
		A (%)	T (%)	G (%)	C (%)	A + T (%)	G + C (%)		
Complete mitogenome	16,021	41.3	38.7	8.4	11.6	80	20	0.032	−0.160
PCGs	11,125	33.9	44.3	11.1	10.7	78.2	21.8	−0.133	0.018
1st codon position	3711	35.0	38.2	16.7	10.1	73.2	26.8	−0.043	0.247
2nd codon position	3707	21.1	47.9	13.4	17.5	69.0	29	−0.388	−0.135
3rd codon position	3707	45.5	46.7	3.2	4.6	92.2	7.8	−0.014	−0.181
tRNA	1432	41.9	38.8	8.7	10.6	80.7	19.3	0.038	−0.098
rRNA	2092	44.6	39.6	5.5	10.3	84.2	15.8	0.059	−0.304
16S rRNA	1280	45.1	38.7	5.7	10.5	83.8	16.2	0.076	−0.296
12S rRNA	812	43.8	41.0	5.2	10.0	84.8	15.2	0.033	−0.316
A + T rich region	1388	44.1	43.2	5.0	7.7	87.3	12.7	0.010	−0.213

**Table 4 genes-14-00414-t004:** Relative synonymous codon usage (RSCU) of *Luperomorpha xanthodera* mitochondrial PCGs. The asterisk (*) represent Termination codon.

Amino Acid	Codon	Count	RSCU	Amino Acid	Codon	Count	RSCU
Phe	UUU	325	1.86	Tyr	UAU	162	1.79
	UUC	24	0.14		UAC	19	0.21
Leu2	UUA	524	5.22	His	CAU	59	1.66
	UUG	21	0.21		CAC	12	0.34
Leu1	CUU	22	0.22	Gln	CAA	57	1.84
	CUC	3	0.03		CAG	5	0.16
	CUA	27	0.27	Asn	AAU	190	1.83
	CUG	5	0.05		AAC	18	0.17
Ile	AUU	391	1.87	Lys	AAA	102	1.87
	AUC	27	0.13		AAG	7	0.13
Met	AUA	241	1.85	Asp	GAU	62	1.8
	AUG	20	0.15		GAC	7	0.13
Val	GUU	66	1.8	Glu	GAA	69	1.86
	GUC	5	0.14		GAG	5	0.14
	GUA	69	1.88	Cys	UGU	30	1.88
	GUG	7	0.19		UGC	2	0.13
Ser2	UCU	107	2.58	Trp	UGA	88	1.91
	UCC	7	0.17		UGG	4	0.09
	UCA	88	2.12	Arg	CGU	19	1.36
	UCG	3	0.07		CGC	1	0.07
Pro	CCU	66	2.06		CGA	34	2.43
	CCC	8	0.25		CGG	2	0.14
	CCA	53	1.66	Ser1	AGU	27	0.56
	CCG	1	0.03		AGC	3	0.07
Thr	ACU	90	2.03		AGA	85	2.05
	ACC	11	0.25		AGG	12	0.29
	ACA	76	1.72	Gly	GGU	50	1.05
	ACG	0	0		GGC	6	0.13
Ala	GCU	63	1.81		GGA	110	2.3
	GCC	16	0.46		GGG	25	0.52
	GCA	59	1.7	*	UAA	0	0
	GCG	1	0.03		UAG	0	0

## Data Availability

The *Luperomorpha xanthodera* sequenced mitogenome was submitted to the GenBank database (ON631248), the multiple sequence alignment used for phylogentic analysis are avalable on FigShare at this weblink: https://doi.org/10.6084/m9.figshare.21957440.v1, accessed on 23 December 2021.

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
