# Peer review of "Characterization of the Complete Mitochondrial Genome of a Flea Beetle Luperomorpha xanthodera (Coleoptera: Chrysomelidae: Galerucinae) and Phylogenetic Analysis"

_genes, 2023, doi:10.3390/genes14020414_

Round 1
Reviewer 1 Report (Previous Reviewer 1)
The manuscript can be published in this form.
Author Response
Dear Dr. Reviewer,
Thank you for your affirmation of my work, and I will continue to make positive contributions to scientific research.
Sincerely,
Jingjing Li
Reviewer 2 Report (New Reviewer)
Dear authors, you have delivered great study to reveal complete mitogenome of Luperomorpha xanthodera, congratulations with the results. The data obtained will be useful fo many directions of biological research, such as phylogeny, taxonomy, biochemistry etc.
The only problem that is usually occurs when taxonomic problem are trying to interpret via genetic study. Under Conclusion you wrote: "Our study identified a close relationship between Luperomopha Weise and Luperini (currently Luperina)". Why do you think so? Are you sure that all representatives of Luperomorpha are possessing the same characters as xantodera to be comparable with species of the Luperini? You know, the genus Luperomorpha Weise, 1887 is defined by the type species Luperomorpha trivialis Weise, 1887 according to International Code of Zoological Nomenclature. Thus, only mitogenome of the type species should have to be studied and compared with other species to declare position of the genus Luperomorpha. The species Luperomorpha xanthodera those mitogenome was studied is not a type species, and could happened that it belongs to other genus, not Luperomorpha. What to do then?
Thank you for your precise work on genetic study of beetles,
wish you successful research in future.
Sincerely,
Author Response
Please see the attachment.

Reviewer 3 Report (New Reviewer)
The article "Characterization of the complete mitochondrial genome of a flea beetle Luperomorpha xanthoderma (Coleoptera: Chrysomeliadae: Galerucinae), and Phylogenetic analysis" presents new genetic data from an interesting organism such as Luperomorpha. All data are presented in a clear and understanding manner. The only flaw which I observed are references that are not mentioned in the main text and the order of references in the main text needs to be checked and corrected, also the manufacturer of the DNA isolation kit needs to be specified.
Author Response
Please see the attachment.

Reviewer 4 Report (New Reviewer)
Brief Summary:
Li et al. provide a new mitochondrial genome of the species Luperomorpha xanthodera. This is a mostly a descriptive study of new mitochondrial data but little effort is made to put into context with other studies and with the evolution of the group, such as the ecology or the evolution of morphological characters. The authors do not emphasize enough on the relevance or connection of their results to other evolutionary studies of Chrysomelidae or to the impact of their new data or results in their field of study. Nevertheless, these new data could be useful for future evolutionary studies of beetles.
General comments:
The methods for mitochondrial genome assembly are well described but some methods of the phylogenetic analyses require more detailed documentation (see my more specific comments below). Additional analyses could also be performed to check for heterogeneous sequence divergence using a software such as ALIGROOVE that is frequently used to detect inflated branch support and spurious sequences that are prevalent in mitochondrial sequence data (Kück et al. 2014, BMC Bioinformatics, 294; López-López and Vogler 2017, Mol. Phyl. Evol., 114:166-174). The datasets supporting the conclusions (i. e., multiple sequence alignments) should also become available to facilitate reproducibility of analyses.
Specific comments:
Line 72: PCG - This abbreviation was not introduced before. I assume it stands for protein-coding genes.
Line 76: Citation is missing for this software
Line 80: Here the abbreviation should be introduced above in the first mention of the word
Line 80: The sequences of amino acids were aligned, not the amino-acids themselves
Line 81: The accuracy of the local alignment algorithm of L-INS-I depends on the data being analyzed and based on which algorithm is being compared to. I would remove the description “highly accurate” unless the authors can justify the high accuracy of the algorithm for their own or other data (or through references).
Line 88: Why did the authors use this specific profile mixture model for the PMSF approach (mtInv+C60+FO+R)? Model selection is not mentioned anywhere in the text. Did the authors compare models using statistical criteria such as BIC or AICc?
Line 90: Change “subjected to” to “analyzed with”
Line 90: There is no mention of of how convergence of the results of the Bayesian analysis was assessed. Did the authors used tracecomp to ensure acceptable effective sizes of the parameters and bpcomp to assess convergence of the topology? This information needs to be added here and also the results.
Line 121: This is nucleotide composition, not combination
Line 152: This sentence is not clear. Please rephrase: which relation did the authors want to investigate?
Line 156: This is Figure 3 not one. Please modify caption of figure3. There are 2 figures currently labeled as figure 1.
Line 167: Change mitogenomes to mitogenome
Line 199: This part fits more to the discussion as it is not strictly a result.
Line 206: Here you say Luperomorpha is sister to Luperina but Luperina is not monophyletic in the tree. Luperomorpha is sister to only a subgroup of Luperina in the ML tree
Line 207: I think the authors do not make correct interpretations of the relationships in the tree. Luperomorpha is clearly not nested in Galerucini
Line 232: The results provide a hint for the phylogenetic position of Luperomorpha but do not provide information about the internal phylogeny of Luperomorpha. Please rephrase.
Figure 3: Remove the word “genus”
Figure 6: The legend shows Galerucina for color yellow but on the tree the word Galerucini is used. Be consistent with terminology of taxonomic groups.
Figures 6 and 7: The phylogentic trees are obviously both rooted with only one species (Chrysomela vigitipunctata) but two species are tagged as outgroup; Please either tag only Chrysomela vigitipunctata as outgroup, or reroot the tree with both Chrysomela aeruginosa and Chrysomela vigitipunctata. Also in which group does Luperomorpha belong to?
Data availability: The multiple sequence alignment used for phylogentic analysis should be provided in a public repository (e.g. Mendeley data, Dryad or figshare).

Author Response
Please see the attachment.

This manuscript is a resubmission of an earlier submission. The following is a list of the peer review reports and author responses from that submission.
Round 1
Reviewer 1 Report
Study of mitogenome seems to be OK. But Galerucini is an extremely rich and diverse group with tangled taxonomy. It is obvious that taxonomic problems in this group can not be resolved with studies of mitogenome. As for me, it would be more interesting to compare standard COX1 sequence with other available bin Gen Bank ones for Luperomorpha to evaluate infraspecific and interspecific variability in the genus.
All specific names in the References are done with capital letters. It must be changed.
Author Response
Dear reviewer,
On behalf of all the contributing authors, I would like to express our sincere appreciations of your constructive comments concerning our article entitled “Characterization of the complete mitochondrial genome of a flea beetle Luperomorpha xanthodera (Coleoptera: Chrysomelidae: Galerucinae), and Phylogenetic analysis” (Manuscript No.: genes-2020970). Please see the attachment.

Reviewer 2 Report
The present manuscript is essentially an isolated characterization of the metagenome of a single well-known species of flea beetles. The species pose no problems in identification or taxonomy. It was a subject of a number of studies because of the recent spreading outside its native range. However, the damage of this species is low and it does not represent an essential quarantine problem. Therefore, the selection of the study taxon is not substantiated and the goals of the research are unclear.
The title of the manuscript does not reflect its content. The manuscript contains mostly various metrics of the mitogenome, but these metrics are not compared to those of closely related species and are not used in the phylogenetic analysis. The phylogenetic analysis is formal and rudimentary and does not advance our knowledge of the evolution of Chrysomelidae in general and Galerucini in particular. It’s based on re-examination of the existing data with insignificant addition of new data.
The detailed characterization of a single mitogenome would be of interest some 10 years ago, but nowadays mitogenomes are routinely reconstructed by standardized methods and this work can be outsourced to commercial companies for reasonable amount of money.
There is no actual discussion. “Discussion” section is rudimentary and does not discuss any of the numerous characteristics of the Luperomorpha xanthodera mitogenome, that constitute the manuscript. There is no comparison of the various metrics of the new mitogenome with the known data, first of all with the mitogenome of Luperomorpha hainana. If there are differences between these species, it is these differences that should be discussed. If there are no differences (except, or course, for barcode regions), there is no need in phylogenetic analysis at generic and higher level, since it would a priori not falsify previous phylogenetic hypotheses (nor can it “support” them!). Chaboo and Clark [45] found nothing about Luperomorpha resemblance to Luperini because they wrote a catalogue of species and not studied relationships of the taxa.
The Conclusions section includes a vague statement that “mitogenome of Luperomopha xanthodera exhibits characteristic sequence structures, gene content, nucleotide composition, codon usage, RNA secondary structures, and PCG evolution rates are similar to those of other Gaerucinae genomes.” It is unclear if these characters are characteristic for Luperomopha xanthodera or for all Gaerucinae, and how they differ from those of other taxa.
This contribution may be of value for a specialized journal like “Mitochondrial DNA” but the “phylogenetic analysis” must be dropped.
Author Response

(The authors gave the same response as above.)

Reviewer 3 Report
The work is interesting and it is always great to have new mitogenome data available in public databases. I congratulate the authors for the effort in generating a new mitochondrial genome. However, the writing of the work is very difficult to follow, the methodology is not very clear and the results and discussion are still quite superficial. I suggest authors to improve these aspects and try to resubmit the work in a new opportunity. Below are some issues that I highlighted in the manuscript.
1) The work needs an intense review of English. It has several parts that are very difficult to understand. In addition to the language, I would also rethink the flow of information to improve clarity.
2) In the line 14: “The nucleotide composition of the mitochondrial genome is 41.3% for adenine (A), 38.7% for thymine (T), 11.6% 14 for guanine (G), and 11.6% for cytosine (C), respectively”. The authors need to review the values according Table 3.
3) In the section 2.2, I couldn't understand what each program was used for. How was MITOZ used, for example? I suggest that the authors try to rewrite the methodologies, with a continuous and understandable flow of steps.
4) In the line 72: “The sequences were aligned in batches using MAFFT software, trimmed using trimAlv1.4”, authors must specify which sequences were used. All PCGs? Nucleotide or amino acids?
5) In the line 75: “To decrease the risk of bias or long-branch attraction due to substitution saturation within species belonging to different genera [19,20], third codon positions were removed from nucleotide-based studies”. Reading the methodology, I didn't understand that the trees were made using the amino acid sequences. I could only understand this in the results. Anyway, what was this alignment with the removal of the third codon position used for? And how was this alignment and removal accomplished?
6) I suggest placing Table 1 in the supplementary material or enriching it with more data. For example, so we can compare other data across genomes, such as percentage GC. With the information it presents, it doesn't add much to the main text.
7) In the line 80: “Using the PCGs_faa dataset in the phylogenetic analysis, maximum likelihood (ML) 80 and Bayesian inference (BI) were used to create phylogenetic trees. The ML analyses 81 were carried out using IQ-TREE V1.6.3 [21] and 1000 ultrafast bootstrap [22] and 1000 82 SHaLRT replicates [23], the PMSF model [24] was used for PCGs_ faa matrix by specify- 83 ing the profile mixture model with the option "-mtInv+C60+FO+R". What is PCGs_faa? Authors should avoid the use of filenames. Are PCGs_faa all the amino acid sequences of PCGs concatenated? Why did the authors decide to do the phylogeny with sequences of amino acids and not nucleotides? Why were the amino acid substitution models for ML and BI different?
8) All the species name must be in italics.
9) In the line 105: The authors need to correct the title of the table 2 and add a legend to explain the acronyms used.
10) The discussion and conclusion are very superficial. The authors may consider adding more data, such as the synteny between the analyzed mitogenomes, to bring more new results to the work.
Author Response

(The authors gave the same response as above.)

Round 2
Reviewer 2 Report
Dear Authors,
I see reasonable revision of your text but these are mostly cosmetic changes. I still think that the phylogenetic analysis, as presented here, should not be included. The detailed characterization of a mitogenome might have a merit on its own, especially if the mitogenomes of the related species were not yet characterized. But proper phylogenetic analysis should be designed to address a specific problem. I see no problem that is resolved by your analysis. I recommend strongly that you omit the phylogenetic analysis and re-submit the mitogenome characterization.
Reviewer 3 Report
I would like to congratulate the authors for the significant improvement in the English correction of the work. However, I am not comfortable with the answers to my questions regarding the methodology. The current methodology is coherent, but the authors did not answer me about the points of greatest doubt. The methodologies that I questioned were removed from the article, which made me insecure about the methodological procedures actually applied. Given the above, I leave the decision on the manuscript to the editor.
